# User Walking Speed and Standing Posture Influence Head/Neck Flexion and Viewing Behavior While Using a Smartphone

**DOI:** 10.3390/healthcare11233027

**Published:** 2023-11-23

**Authors:** Yi-Lang Chen, Hong-Tam Nguyen

**Affiliations:** 1Department of Industrial Engineering and Management, Ming Chi University of Technology, New Taipei 243303, Taiwan; m05218010@mail2.mcut.edu.tw; 2Buy2sell Vietnam, Ho Chi Minh 751000, Vietnam

**Keywords:** smartphone use, walking speed, head/neck flexion, gaze angle, viewing distance

## Abstract

The habit of using smartphones while walking has become widespread among modern individuals, particularly when pedestrians are in a hurry. However, there has been little exploration into the differences between standing and walking at various speeds in terms of smartphone use. In this study, we examined 60 young participants (30 men and 30 women) who engaged in smartphone tasks such as one-handed browsing or two-handed texting while standing, walking slowly, and walking normally. The measured variables included neck flexion (NF), head flexion (HF), gaze angle (GA), and viewing distance (VD). The study findings indicate that using smartphones while walking may cause a more pronounced kyphotic curve in the cervical spine compared to when standing, leading to increased strain in the neck region. The heightened neck load can be attributed to the concurrent dynamic nature of both walking and smartphone usage. Moreover, two-handed texting had a more detrimental impact on NF, HF, and GA when contrasted with one-handed browsing. The interplay among hand operation, posture, and maintaining arm position displayed an uncertain correlation with VD. While women typically exhibited smaller NF, HF, and GA than men, it is important to explore whether their shorter VD might contribute to increased eyestrain.

## 1. Introduction

In today’s world, smartphones serve as gateways to the internet, acting as portable computers that facilitate information exchange, support e-commerce, enhance social interactions, and offer personal entertainment. These aspects have become integral components of modern life [1]. While smartphones offer numerous advantages, a variety of studies have emphasized that their usage can pose hazards [2]. These issues, combined with dedicated efforts to find solutions, are increasingly gaining attention as significant concerns. Within the field of ergonomics, the forefront is occupied by the imperative task of addressing excessive smartphone usage that negatively impacts the body.

Due to the inherent convenience of mobile phone usage, it finds applications in a diverse range of activities, including varying postures (standing or walking) and hand operations (one-handed browsing or two-handed texting). These activities introduce different physical demands on users, with a notable focus on neck and head flexion (NF and HF, respectively). In particular, the changes in NF and HF that occur during smartphone use have undergone thorough investigation in the literature [3,4,5]. A larger NF is generally associated with heightened strain on the cervical spine [6], consequently escalating the risk of neck and shoulder injuries [7]. For those who frequently use smartphones, the prevalent condition recognized as text-neck syndrome has been observed [3,8]. However, the concurrent head and neck movement when a text-neck posture is exhibited is complicated. NF typically encompasses the forward flexion of both the head and the cervical spine as a unified unit, while HF refers to head bending with the upper cervical spine acting as the axis of rotation. Indeed, the mechanism of forward head and neck flexion varies fundamentally with posture [9]. Research also demonstrated that two-handed texting engendered larger NF compared to one-handed browsing, and this divergence is primarily attributed to distinct visual demands among tasks [3,4,10].

The findings also uncovered that, despite exhibiting lower NF during dynamic walking while using a smartphone compared to stationary standing, there is noticeable activity within the associated neck muscle groups, regardless of whether the dominant or non-dominant hand is in use [3]. This phenomenon arises from the oscillating motions of the head, which generate both linear and angular accelerations, necessitating continuous engagement of the neck muscles [11]. In a study by Han and Shin [4], it was emphasized that smartphone users display more pronounced NF and HF angles while walking, in contrast to non-smartphone users. Similar outcomes were also observed by Yuan et al. [12] in the standing posture. Likewise, Kim et al. [13] observed that using smartphones while walking resulted in changed gait features, including reduced cadence and velocity, adjusted stride length and gait cycle, as well as prolonged ground contact time. This finding suggests that when investigating smartphone use during walking, it becomes imperative to account for its dynamic nature, including visual behavior in dynamic environments.

In addition to the potential musculoskeletal injuries arising from excessive forward NF, prolonged smartphone use also contributes to eyestrain [12,14,15,16,17]. As highlighted by Choi et al. [18], participants with dry eye symptoms not only demonstrated statistically significant variations in smartphone usage time but also a notable correlation between the Ocular Surface Disease Index and text-neck syndrome. However, a research gap exists in terms of assessing the ergonomic implications of visual strain during smartphone usage. While previous studies have incorporated GA and VD alongside NF and HF angles [19,20,21], a comprehensive investigation into the visual strain associated with smartphone use across different usage scenarios is still lacking.

The sight of individuals using smartphones while walking has become widespread, even during busy activities when people are absorbed in their phone screens. In previous experiments involving smartphone use during walking, many required participants to maintain a preferred pace [3,7]. Although using smartphones while walking has been linked to a decrease in cadence and velocity [7,12], certain situations arise where smartphone users maintain their regular walking speed. Our observations indicate that in public spaces, such as when nearing a subway station or swiftly crossing a road, smartphone users do not consistently reduce their walking speed. In specific instances, maintaining nearly normal walking speed is essential due to time limitations. Moreover, prior investigations into smartphone use while walking mainly centered on assessing the load on the cervical spine [3,7,10], with limited exploration into eyestrain. Nevertheless, the visual load experienced while walking might vary due to dynamic effects, setting it apart from typical static postures like standing or sitting.

In light of the details mentioned earlier, the aim of this study was to elucidate the impact of using a smartphone with either one or both hands on head and neck posture, as well as visual characteristics, during static standing and various walking speeds. Our hypothesis posited that engaging in a dynamic walking posture and performing a two-handed texting task would lead to heightened strain on the neck. Furthermore, we anticipated that the effects of diverse walking speeds might interact with the responses elicited by varying methods of hand manipulation.

## 2. Methods and Materials

### 2.1. Participants

A total of 60 young individuals who regularly use smartphones, and who had no history of musculoskeletal disorders or visual impairments, were enrolled in this study. Recruitment was carried out through announcements on the bulletin board of Ming Chi University of Technology (New Taipei, Taiwan) during 1–20 March 2023. The volunteers were then interviewed and informed of the details of the test procedure. To align with the experimental design of this study, a sex-balanced sample of 30 men and 30 women was utilized. All participants possessed corrected normal vision. Adhering to established research standards, inclusion criteria mandated a minimum of 1 year of smartphone usage with a daily duration of at least 3 h. Unqualified participants were excluded during the recruitment phase. All participants provided informed consent. The experiment adhered to the 2013 World Medical Association Declaration of Helsinki and received ethical approval from the Ethics Committee of National Taiwan University (code: NTU-REC-2020-12EM-025).

### 2.2. Posture Measurement

Figure 1 depicts the four variables: NF, HF, GA, and VD. To obtain measurements, we captured symmetrical sagittal photographs using the MacReflex motion capture system (Qualisys, Gothenburg, Sweden). To facilitate measurements, we strategically placed four adhesive reflective markers on specific anatomical points: the tragus (T), canthus (C), seventh cervical spinous process (C7), and seventh thoracic spinous process (T7). Additionally, a marker was affixed to the middle point of the phone’s length. The arrangement of these markers and their corresponding angles and distances during smartphone use is presented in Figure 1. During data collection, participants engaged in standing and walking activities at two predetermined speeds (slow and normal) on a treadmill (CS-5728, Chanson, Taipei, Taiwan). The smartphone tasks included one-handed browsing and two-handed texting. Notably, for accurate NF measurements, previous research utilized the upper thoracic angle (UTA) as a reference due to the need to consider forward trunk bending [3,18]. The UTA is defined as the angle between the line connecting C7 and T7 and the vertical line. Moreover, actual VD data were determined by normalizing half of the phone length, ensuring consistency across measurements.

### 2.3. Design and Procedure

The study encompassed a total of 18 trial sessions, with the primary aim of gathering data related to NF, HF, GA, and VD, with a specific focus on smartphone usage. During these sessions, participants assumed three different postures and performed two types of hand operations, with each combination repeated three times. For each variable, we calculated the mean of the two closest measurements from the three repetitions for subsequent analysis. Data were collected from participants while they were in a stationary stance and while walking at their perceived slow and normal speeds. The two speeds were preselected based on subjective assessments made by each participant. All participants were directed to identify their slow (or normal) walking speed during a 2 min assessment on a treadmill, simulating their usual walking pace. The recorded speeds were subsequently employed in the ensuing tests. The mean slow and normal speeds determined by the participants prior to the test were 2.04 and 3.95 km/h, respectively. Throughout each trial, participants were allowed to use their own smartphones. For the one-handed usage scenario, participants viewed randomly selected videos. Conversely, in the two-handed usage condition, participants sequentially responded to predefined questions transmitted via LINE (Z Holdings, Tokyo, Japan). These questions covered various everyday topics such as weather updates and recent noteworthy news, as well as basic information (sex, age, stature, and body weight). During both tasks, participants were instructed to hold the phone in their preferred way, without any restrictions on its orientation relative to their head, consistent with the approach employed by Yoon et al. [3].

During the experiment, each testing trial lasted for 1.5 min, and within the final 30 s interval, four sets of measurements were taken [22], with readings captured at 10 s intervals. The recorded values from these measurements were then averaged to generate representative data for subsequent analysis. To ensure accuracy and participant comfort, a rest period of at least 2 min was implemented between consecutive trials. During these breaks, participants were instructed to relax in a chair with neck support, promoting complete relaxation. Each participant completed three testing sessions across three half-day periods in total.

### 2.4. Statistical Analysis

The data collected in this study underwent comprehensive analysis using SPSS 23.0 statistical software (IBM Corp., Armonk, NY, USA). A significance level of 0.05 was applied to all statistical tests. The primary objective was to explore how participant sex (30 men and 30 women), posture (standing, slow walking, and normal walking), and hand operation (one-handed browsing and two-handed texting) influenced the measured variables (NF, HF, GA, and VD). This study was conducted through a three-way repeated-measures analysis of variance (ANOVA). To identify significant differences among different variable levels, post hoc comparisons were carried out using Tukey’s test or an independent t test. Before conducting the analyses, the alignment of numerical variables with a normal distribution was assessed using the Kolmogorov–Smirnov test. Additionally, Levene’s test was employed to examine the equality of variances, ensuring the robustness and reliability of the analytical framework.

## 3. Results

In our study, sixty young men and women were recruited as participants. Their main demographic details are presented in Table 1. Among the men, the mean (standard deviation) age was 21.7 (1.8) years, mean height was 173.8 (6.8) cm, and mean body mass was 70.4 (5.0) kg, whereas those of the women were 22.4 (2.4) years, 160.0 (6.1) cm, and 54.7 (4.9) kg, respectively. An independent test revealed no significant age difference between the two participant sex groups.

The results of the three-way ANOVA are summarized in Table 2, outlining how participant sex, hand operation, and posture functioned as independent variables. The analysis revealed significant impacts of sex on all responses (all *p* < 0.001). While hand operation had no influence on VD, it showed an influence on NF, HF, and GA (all *p* < 0.05). Additionally, posture had a significant effect solely on NF (*p* < 0.001).

The main effects and the outcomes of the independent t test (or Tukey’s test) for paired (or multiple) comparisons related to the participant sex, hand operation, and posture variables are illustrated in Figure 2, Figure 3 and Figure 4. Figure 2 depicts that women displayed lower values across all four measured responses compared to men (all *p* < 0.001). In Figure 3, it is evident that the two-handed task resulted in larger NF (*p* < 0.001), HF (*p* < 0.05), and GA (*p* < 0.05) values compared to the one-handed task. Discrepancies were observed in NF (*p* < 0.001) between the standing and walking postures, while no difference was found between the two walking speeds, as illustrated in Figure 4.

It is important to emphasize that the interaction between hand operation and posture had a significant impact on VD (*p* < 0.01). The comprehensive analysis outcome is illustrated in Figure 5. Notably, in a standing posture, the VD linked with one-handed operation was higher than that of two-handed operation (*p* < 0.05). However, an opposite trend was observed when the participant walked at their normal speed (*p* < 0.01). This intriguing interplay between hand operation and posture highlights the complex and nuanced dynamics influencing the study outcomes.

## 4. Discussion

This study systematically investigated variations in head/neck posture (HF, NF) and viewing behavior (GA, VD) between men and women during smartphone usage while standing and walking, encompassing both one-handed and two-handed tasks. The examination covered both stationary standing and dynamic walking at slow and normal speeds. While the expected outcomes were consistent with previous research in terms of the impact of smartphone use on standing posture, this study brought to light an interesting interplay between different walking speeds and hand operations, particularly evident in the case of VD during walking. Furthermore, the effects on NF and HF demonstrated inconsistencies between standing and walking situations (Figure 4). The increased kyphotic cervical posture observed during walking may potentially result in greater strain on the neck muscles. This atypical posture could be attributed to the dual demands of simultaneously paying attention to both the environment around and the smartphone screen while walking.

Historically, no prior investigations have examined the impact of smartphone use on neck/head forward flexion and visual behaviors during different walking speeds. The present study unveiled a significant difference in mean NF (37.7°) while standing, in contrast to the measurements of 31.7° and 32.1° recorded during walking at slow and normal speeds, respectively. Interestingly, the difference was consistent across the two walking speeds, suggesting that the walking pace itself may not play a significant role. These outcomes align somewhat with previous study findings. While Yoon et al. [3] and Han and Shin [4] conducted their experiments in dynamic walking scenarios on real roads and treadmills, respectively, our study employed a treadmill to simulate walking conditions. Despite the simulated environment, our participants reported heightened awareness of their surroundings and an increased focus on maintaining stable walking during the experiments. It is worth noting that Yoon et al. [3] based their study on participants’ individually preferred speeds, without explicitly providing speed data. On the other hand, Han and Shin [4] observed a mean preferred walking speed of about 3.65 km/h (with mean speeds of 3.82 km/h for browsing and 3.49 km/h for texting), slightly slower than the normal pace (mean 3.95 km/h) observed in our participants. However, the difference in NF between postures was evident not only at the normal speed but also at the slow speed (mean 2.04 km/h). This observation implies that the nature of dynamic movements might have a more significant impact on NF than mere variations in walking speeds.

While smartphone usage in a standing posture exhibited higher NF, it is important to clarify that this does not necessarily equate to a heavier neck load compared to smartphone usage while walking. Interestingly, the head posture during walking did not show a concurrent reduction with NF, along with no significant differences in HF across the three postures, as depicted in Figure 4. This observation might be attributed to the pronounced forward head posture [23] induced by smartphone usage while walking. Our study further unveiled that while walking and using a smartphone, participants exhibited relatively smaller cervical spine flexion and greater HF. This unnatural posture may contribute to increased kyphosis in the cervical region (e.g., upper cervical), consequently leading to higher levels of muscle activity in the neck. This finding may elucidate why the mean activation of neck extensor muscles while using a phone during walking was approximately 40% higher compared to when standing, as observed in Yoon et al.’s study [3].

Previous findings have consistently demonstrated that texting with two hands induces higher NF or HF compared to browsing with one hand, regardless of whether the individual is standing or walking [3,4]. This current study also yielded similar results (Figure 3). On the contrary, a recent study conducted by Brühl et al. [24] reported that the measurement outcomes were not affected by the standing or walking conditions. Various experimental settings may yield divergent study results, and it is important to take note of these differences when making comparisons.

The ANOVA results highlighted a significant interaction effect between hand operation and posture on VD. As depicted in Figure 5, the VDs for two hand operations during standing and normal walking exhibited an opposing trend when using a smartphone, with no discernible difference in VD during slow walking. The increasing prevalence of smartphone use has sparked concerns about ophthalmic issues arising from prolonged usage [25] and relatively short VD [20,26]. Bababekova et al. [27] reported that the mean VDs for reading messages and internet browsing on smartphones were 36.2 cm and 32.2 cm, respectively. These distances notably fall short of typical VDs for other electronic devices [28]. The act of viewing a screen at such close VD intensifies visual demands like accommodation and vergence, potentially leading to an increase in near-point stress [29,30], elevated intraocular pressure [31], and exacerbated symptoms of asthenopia [32]. Ironically, holding the phone closer due to visual strain is counterproductive because the shorter VD intensifies visual demands [32]. However, alterations in VD during smartphone use in stationary and dynamic postures have not been examined until now. In this study, during standing and smartphone use, two-handed operation exhibited a smaller VD (30.5 cm). When walking at slow and normal speeds, the VDs were 33.1 cm (*p* < 0.05) and 33.4 cm (*p* < 0.01), respectively. Bimanual manipulation during dynamic walking increased VD, possibly due to a smaller NF. Interestingly, we observed that dynamic body movements might lead to imbalances in one-handed holding, while participants adopted a stiffer elbow position to avoid fatigue (with a smaller elbow angle and the upper arm positioned closer to the lateral body). This resulted in a shorter VD. In contrast, bimanual manipulation did not exhibit this phenomenon. As suggested by Ralph M. Barnes in the 1930s, arms should be symmetrical and in opposite directions and movements should be made simultaneously [33]. This approach may be more realistic when performing one-handed browsing tasks while walking. This led to a reduced VD, which could potentially escalate the visual load in this context. Furthermore, Merbah et al. [34] discovered that there was no discernible correlation between the distance from the head to the smartphone and the chosen postural strategy. Their finding prompts us to acknowledge that the positioning of the device is influenced by factors associated with the upper limb. However, comprehensive factors such as NF, hand holding posture stress, and visual comfort likely collectively determine VD when using a smartphone during walking. This could lead to excessive visual load, warranting further investigation.

The study findings also indicated a significant influence of participant sex on all measured responses (Figure 2). Women tended to exhibit smaller NF, HF, and GA when using smartphones. While these variations could potentially be influenced by inherent anthropometric differences between sexes, such as body height and arm length, it is more likely that men and women have distinct postural characteristics. Notably, Brink et al. [35] found no substantial correlation between body angle measurements and body height within either sex. In a similar vein, Guan et al. [19] observed that during smartphone use while standing, the difference in NF between sexes was 4.8° (*p* < 0.001, compared to 5.2° in this study). However, there was no significant difference in NF and HF when not using a smartphone (within 0.8°). This result is supported by the findings of Korakakis et al. [36], who noted that women adopted more erect postures compared to men when instructed to sit with their optimal postures. This phenomenon was also evident and more pronounced in the standing and walking postures of this study. The smaller VD observed in women compared to men may not solely be attributed to different postures but is primarily a consequence of the relatively shorter forearm length in females. This shorter forearm length led to a reduced VD in comparison to men [37]. Whether this variation could potentially result in excessive visual strain for women requires further investigation for clarification.

This study presents several limitations. Firstly, the sample size consisted solely of young men and women; thus, the findings may not be directly applicable to other populations, such as children and the elderly. Additionally, the short duration of smartphone usage (1.5 min) in this study contrasts with real-life daily usage, potentially limiting the generalizability of the results to practical scenarios. Another limitation is that the participants’ subjectively determined slow and normal walking speeds (2.04 and 3.95 km/h, respectively) were employed, which slightly differed from those used in previous studies. Furthermore, while VD was used as an index of visual load, it is important to acknowledge that it is an indirect measure. Future research could benefit from verifying the influence of VD through direct measurements, such as critical flicker fusion frequency, to provide more comprehensive insights.

## 5. Conclusions

This study conducted a preliminary exploration into the disparities in head/neck posture and viewing behaviors of smartphone users across three postures: standing and two walking speeds (slow and normal). The findings demonstrated that, irrespective of walking speed, using a smartphone while walking led to a more pronounced disparity between the cervical and head positions, resulting in increased strain on the neck compared to smartphone use while standing. The heightened neck load can be attributed to the concurrent dynamic nature of both walking and smartphone usage. Furthermore, the sustained arm posture required to hold smartphones might directly impact NF and VD. This study also highlighted distinct sex-based differences in the responses. While some of these differences could be attributed to inherent sex-based variations, it remains to be determined whether the smaller VD observed in women leads to higher eyestrain, which requires further confirmation. Our findings suggest that it is advisable to limit smartphone usage when walking, and particularly when employing two-handed operation.

## Figures and Tables

**Figure 1 healthcare-11-03027-f001:**
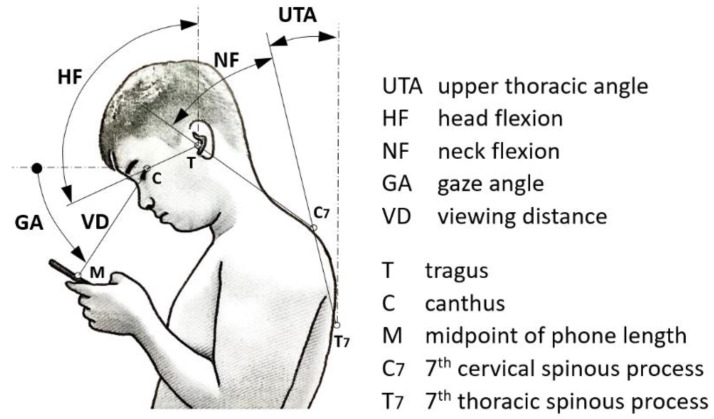
Schematic of the markers and definitions of angles and distance of human body.

**Figure 2 healthcare-11-03027-f002:**
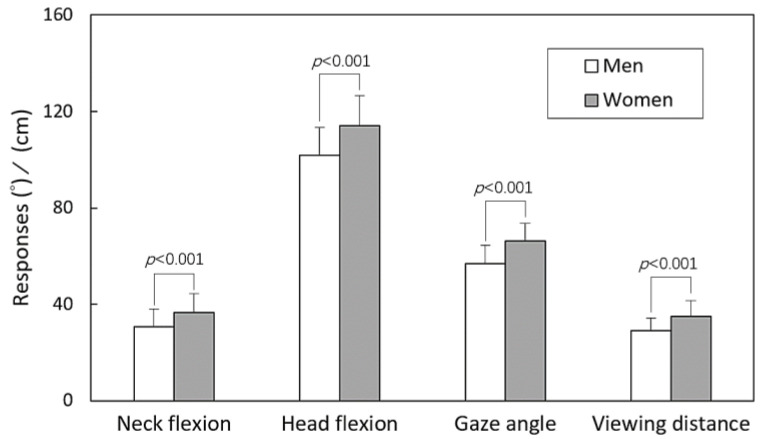
Comparisons of main effects across sexes using independent t test for all responses.

**Figure 3 healthcare-11-03027-f003:**
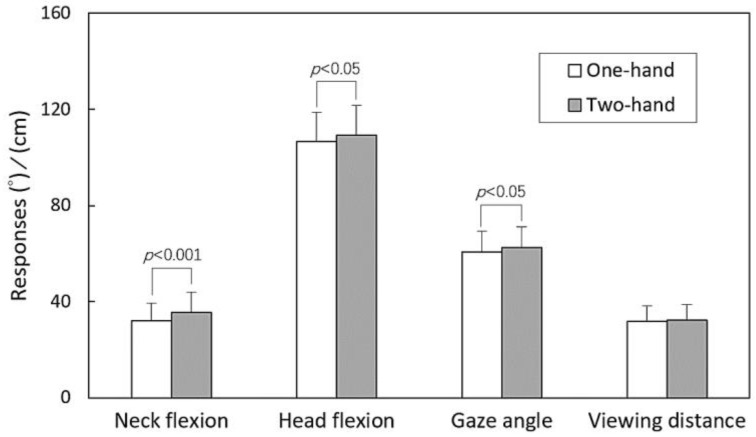
Comparisons of main effects across hand operations using independent t test for all responses.

**Figure 4 healthcare-11-03027-f004:**
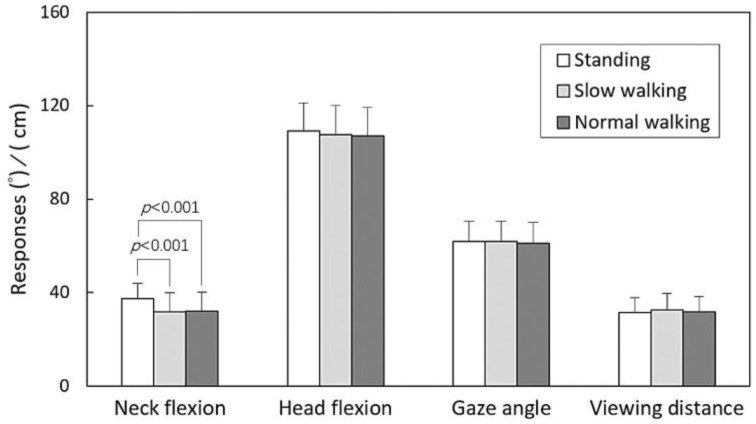
Comparisons of main effects across three postures using Tukey’s test for all responses.

**Figure 5 healthcare-11-03027-f005:**
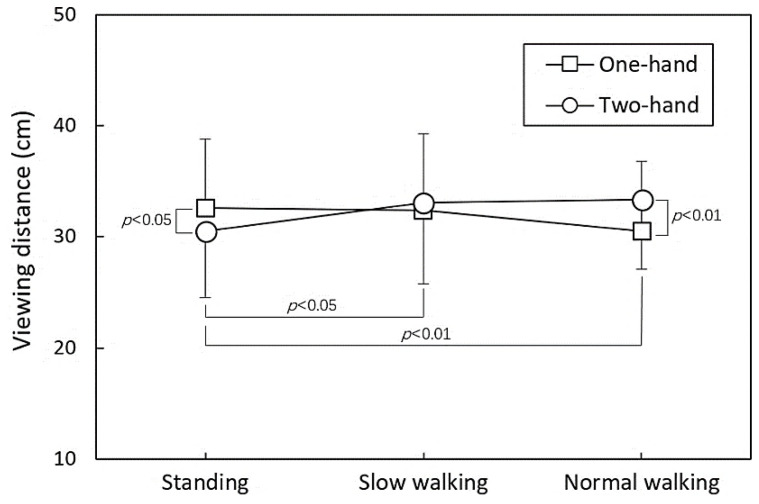
Comparisons of viewing distance between hand operation within three postures.

**Table 1 healthcare-11-03027-t001:** Anthropometric data for the sixty participants included in the study.

	Men (n = 30)		Women (n = 30)	
Items	Mean	Standard Deviation	Mean	Standard Deviation
Age (years)	21.7	1.8	22.4	2.4
Height (cm)	173.8	6.8	160.0	6.1
Body mass (kg)	70.4	5.0	54.7	4.9

**Table 2 healthcare-11-03027-t002:** Results of the three-way analysis of variance for four measured responses.

Variables	Responses	SS	DF	MS	F	*p*
Sex	Neck flexion	2915	1	2915	58.7	<0.001
	Head flexion	13,827	1	13,827	123.6	<0.001
	Gaze angle	7660	1	7660	136.8	<0.001
	Viewing distance	3326	1	3326	97.6	<0.001
Hand operation	Neck flexion	1061	1	1061	21.4	<0.001
	Head flexion	736	1	736	6.6	<0.05
	Gaze angle	347	1	347	6.2	<0.05
	Viewing distance	23	1	23	0.7	0.408
Posture	Neck flexion	2293	2	1147	23.1	<0.001
	Head flexion	334	2	167	1.5	0.226
	Gaze angle	30	2	15	0.3	0.764
	Viewing distance	87	2	43	1.3	0.281
Sex × hand operation	Neck flexion	1	1	1	<0.1	0.905
	Head flexion	21	1	21	0.2	0.668
	Gaze angle	67	1	67	1.2	0.275
	Viewing distance	20	1	20	0.6	0.440
Sex × posture	Neck flexion	38	2	19	0.4	0.684
	Head flexion	71	2	35	0.3	0.728
	Gaze angle	10	2	5	0.1	0.918
	Viewing distance	24	2	12	0.4	0.705
Hand operation × posture	Neck flexion	26	2	13	0.3	0.773
	Head flexion	334	2	167	1.5	0.226
	Gaze angle	18	2	9	0.2	0.851
	Viewing distance	371	2	185	5.4	<0.01
Sex × hand operation × posture	Neck flexion	18	2	9	0.2	0.837
	Head flexion	8	2	4	<0.1	0.964
	Gaze angle	71	2	35	0.6	0.532
	Viewing distance	7	2	4	0.1	0.902

Notes: SS, sum of square; DF, degree of freedom; MS, mean square.

## Data Availability

The data are available upon reasonable request to the Corresponding Author.

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
