# Peer review of "User Walking Speed and Standing Posture Influence Head/Neck Flexion and Viewing Behavior While Using a Smartphone"

_healthcare, 2023, doi:10.3390/healthcare11233027_

Round 1

Reviewer 1 Report

Comments and Suggestions for Authors

Dear authors,

Thank you for the opportunity to review this manuscript. This study aimed to examine a number of selected variables (neck flexion, head flexion, gaze angle, and viewing distance) while performing different smartphone tasks. 

Introduction

The Introduction is well-written and provides a good background on the topic. I recommend the authors to rephrase the last paragraph. The number of participants and the study’s design are described in the methods section. Thus, I recommend the authors use this last paragraph simply to state the study’s objectives and the hypothesis defined. 

Methods

During the text, the authors mentioned the “average” age, height, etc. In the Table 1, the “mean” term is used. I recommend using the “mean” instead of “average”, and consistency during writing. 

Can you provide more details regarding the participants’ selection? Where did it occur? Did you aim to have a sample balanced in terms of gender or this was only a coincidence? There were participants excluded during the recruitment? 

Line 136-137: Can you explain better how was made the subjective assessment of the two speeds? 

Line 146-147: Was this protocol based on previous literature? 

Results

In Table 2, please include the meaning of the abbreviations in the footnote. 

Discussion

Line 249: “doesn’t”, please use “does not”. 

Conclusion

Please provide some practical applications based on your results. 

Reviewer 2 Report

Comments and Suggestions for Authors

Dear all, I thank you for the opportunity to read and review this study. Here are some considerations to broaden the understanding of the manuscript:

Summary

1.

Introduction

I consider that the main facts have been presented. The text is easy for the reader to read. Finally, the theoretical basis is quality, as well as the justification for carrying out the study was well prepared.

Methodology

1. The information in Table 1 must be presented as results and not in this section

Results

1. In general, throughout the text I suggest replacing the word "gender" with "sex". Furthermore, also replace "male and female" with "man and woman".

2. It is important that two procedures are carried out in this section:

a) Not just writing, for example: The results were presented in Table 1, or the results were presented in Figures 1, 2, and 3. But present the findings objectively! For example: Figure 2 shows XXXX. Comparatively, men indicated XXXX in the following variables, while men indicated XXX.

b) Therefore, I suggest including some text above each Table and Figure. Therefore, avoid presenting Figures without text

Discussion

I consider this section to be rigorous in organizing the facts. Thus, it also offers limitations and suggestions for future studies.

Conclusion

It's well structured!

Comments on the Quality of English Language

Fine

Reviewer 3 Report

Comments and Suggestions for Authors

Excellent work! 

I have but two suggestions:

1. You offer a more detailed explanation of NF and HF in the discussion section. I, as the reader, wondered about the details of this difference throughout the document. Please consider moving this explanation earlier in the paper. In the Introduction perhaps. 

2. I believe you do not need to include the Power column in Table 2. 

Otherwise, this is a very well written manuscript. 
